# *Drosophila* DAxud1 Has a Repressive Transcription Activity on *Hsp70* and Other Heat Shock Genes

**DOI:** 10.3390/ijms24087485

**Published:** 2023-04-19

**Authors:** Jorge Zuñiga-Hernandez, Claudio Meneses, Macarena Bastias, Miguel L. Allende, Alvaro Glavic

**Affiliations:** 1Millennium Institute Center for Genome Regulation (CGR), Department of Biology, Faculty of Sciences, University of Chile, Santiago 7800003, Chile; jomzuniga@ug.uchile.cl (J.Z.-H.);; 2Millennium Nucleus Development of Super Adaptable Plants (MN-SAP), Santiago 8331150, Chile; 3Departamento de Genética Molecular y Microbiología, Facultad de Ciencias Biológicas, Pontificia Universidad Católica de Chile, Santiago 8331150, Chile; 4Centro de Biotecnología vegetal, Facultad de Ciencias de la Vida, Universidad Andrés Bello, Santiago 8370035, Chile

**Keywords:** CSRNP, DAxud1, transcription repression, *hsp70*

## Abstract

*Drosophila melanogaster* DAxud1 is a transcription factor that belongs to the Cysteine Serine Rich Nuclear Protein (CSRNP) family, conserved in metazoans, with a transcriptional transactivation activity. According to previous studies, this protein promotes apoptosis and Wnt signaling-mediated neural crest differentiation in vertebrates. However, no analysis has been conducted to determine what other genes it might control, especially in connection with cell survival and apoptosis. To partly answer this question, this work analyzes the role of *Drosophila* DAxud1 using Targeted-DamID-seq (TaDa-seq), which allows whole genome screening to determine in which regions it is most frequently found. This analysis confirmed the presence of DAxud1 in groups of pro-apoptotic and Wnt pathway genes, as previously described; furthermore, stress resistance genes that coding heat shock protein (HSP) family genes were found as *hsp70*, *hsp67*, and *hsp26*. The enrichment of DAxud1 also identified a DNA-binding motif (AYATACATAYATA) that is frequently found in the promoters of these genes. Surprisingly, the following analyses demonstrated that DAxud1 exerts a repressive role on these genes, which are necessary for cell survival. This is coupled with the pro-apoptotic and cell cycle arrest roles of DAxud1, in which repression of *hsp70* complements the maintenance of tissue homeostasis through cell survival modulation.

## 1. Introduction

Tissue homeostasis is the process through which tissues are maintained in a stable and balanced state. It is an important task of biological organisms to sustain a proper physiological function, which is achieved through the regulation of cell proliferation, cell death, and cell differentiation [1]. The regulation of these cell processes is achieved through a variety of mechanisms including cell signaling, metabolism, and gene expression [2]. Cell proliferation and cell death mediated by apoptosis are opposite processes that keep a balanced cell number during development and adulthood [3]. In addition, gene expression is an important factor in tissue homeostasis, in which genes are turned on or off to produce proteins that are necessary for cell proliferation, death, and differentiation [2]. Changes in gene expression can lead to changes in cell behavior, which can affect tissue homeostasis to adapt to both physiological and environmental changes that could impair tissue and cell integrity. In certain environmental contexts (e.g., heat and oxidative stress), cells can rapidly synthesize chaperone molecules such as heat shock proteins (HSPs) through a rapid transcriptional response [4]. HSPs are a superfamily of molecular chaperones that help to preserve the structural and functional integrity of proteins in the cell and promote survival under various stress conditions. The main families of HSPs include the HSP90, HSP70, HSP60, and small HSP families, classified by their molecular weight [5]. HSP90, the most abundant and highly conserved family of HSPs, functions as a molecular chaperone, assisting in the proper folding and stabilization of a wide range of proteins [6]. HSP70 family proteins also function as molecular chaperones, assisting in protein folding and preventing protein aggregation. HSP70 is also involved in designating misfolded proteins for degradation [7]. These proteins play a crucial role in maintaining cellular homeostasis by assisting in protein folding, preventing protein aggregation, and facilitating protein degradation. In *Drosophila*, there are 5 *hsp70* gene paralogs whose promoters and expression have been thoroughly characterized [8]. HSP60 family proteins are involved in protein folding within mitochondria, where they assist in the folding of mitochondrial proteins [5]. Lastly, the small HSP family consists of small, highly conserved proteins that function as molecular chaperones and are involved in preventing protein aggregation and protecting cells from stress-induced damage [9].

As previously mentioned, HSP chaperones act together to protect proteins from damage and degradation and are essential for cell viability and normal cellular function, not only maintaining proteostasis, but also inhibiting apoptosis [7,10,11]. In animals, *hsp* gene transcriptional induction is very fast, with *hsp70* mRNA significantly increasing in less than 5 min upon heat shock [10]. Stress conditions such as heat shock induce recruitment and activation of transcriptional machinery on the *hsp70* promoter, this heat shock response is mainly commanded by HSF (Heat Shock Factor), a canonical transcriptional activator for heat shock response, which induces transcription of *hsp70*, *hsp60*, *hsp67*, and small HSPs (*hsp27*, *hsp26*) when environmental temperature raises even just few degrees over physiological threshold [5]. However, the chromatin-associated proteins vary for other stress conditions, and the diversity and factor combinations that determine a selective adaptation are under constant research, with researchers advancing questions and proposals about chromatin factors that can participate in both induction and repression [12].

The adequate regulation of the expression of these genes and their products is key to tissue homeostasis since although these molecular chaperones protect against certain levels of stress, prolonged overexpression can lead to the survival of damaged cells, preventing their entry into apoptosis and renewal, thus favoring phenomena such as cancer [13]. In fact, many cancers considered aggressive have high levels of heat shock proteins such as HSP70, which are under study as a target for treatment [13,14,15,16]. Due to this effect on cell and tissue homeostasis, gene repression mechanisms exist to prevent expression when it is not required and preserve the equilibria between cell proliferation, mitotic arrest, and apoptosis.

Gene repression, an important aspect of gene expression regulation, allows cells to prevent a gene or set of genes from being expressed. Gene repression is essential for normal cellular processes, including development, homeostasis, and differentiation [17]. This is achieved by means of a set of factors distributed in chromatin that possess a range of molecular functions such as histone modifications and post-translational modifications of transcription factors [18]. With these mechanisms among other, cells can regulate and maintain the levels of certain proteins and metabolites to ensure their proper functioning. Thus, the expression of HSP genes must be inactivated when they are not required since their overexpression leads to excessive cell survival in a tissue, preventing an appropriate renewal of defective cells [14,15,16,19]; furthermore, cells in the growth phase are affected by the expression of *hsp70* since their biosynthesis consumes the available nitrogen, which means that *hsp70* overexpression is especially disadvantageous during development [20]. Due to these effects, tight control of the expression of chaperone proteins such as *hsp70* is necessary; plus, the cell has mechanisms that maintain the state of lower expression under non-stressful conditions [4], both at transcriptional and post-translational levels. In this context, it is necessary to search for new key regulators to understand the expression of *hsp70* and other *genes* from its superfamily.

The *dAxud1* gene is the *Drosophila* ortholog of the mammal CSRNP protein family (Cysteine Serine Rich Nuclear Proteins) [21,22,23,24]. This protein family, which is conserved in metazoans, has been suggested as putative transcription factors with putative roles in promoting apoptosis [24] and neural development [23,25]. Early studies proposed a function for the human ortholog as a tumor suppressor in different tissues [21]. Moreover, analysis of the *Drosophila* ortholog revealed that its proapoptotic role is dependent on the JNK pathway [24]. Besides this, multiple studies about mammal orthologs indicate a convincing correlation of the CSRNP family with stress responses, as their mRNAs expression rises in cells and tissues exposed to different stressful stimuli, including oxidative stress [26,27], bacterial infection [28], physical pressure [29], and metabolic stress by sprint running activity in skeletal muscle [30]. This scrutiny suggests that the next analysis of this protein family should analyze the molecular roles of the dynamics of the adaptive and non-adaptive functions of DAxud1 during the stress response and their implications for apoptosis and cell survival during various stress conditions. Here, a whole-genome occupancy analysis (TaDa-seq) was performed, and this revealed that DAxud1 has a connection to HSP expression due to its possible repressive effect on the expression of *hsp70* and other genes from its family such as *hsp26* and *hsp67*. This repressive effect on these genes could be related to a reduction in cell survival due to the role of DAxud1 in maintaining tissue homeostasis (eliminating damaged cells) and eventually regulating over-proliferation. This function of DAxud1—repressing genes that promote cell survival—is complementary to its previously described pro-apoptotic function, so its role in tissue homeostasis is likely to be broader, controlling genes with contrasting functions in opposite ways.

## 2. Results

### 2.1. Distribution of DAxud1 over the Drosophila Melanogaster Genome

In earlier studies, vertebrate and invertebrate DAxud1 orthologs have transcription factor-like attributes, including DNA binding as well as acidic and transcription transactivation domains in the C-terminal tail [22,24,25]. Furthermore, DAxud1 in both vertebrates and invertebrates have been associated with stress response and tumor suppression through the activation of apoptosis and cell cycle inhibition [21,24,27]; although, until now, these processes have not been linked to the putative transcriptional function of DAxud1. Based on this information, our initial focus was to determine which groups or categories of genes are transcriptionally regulated and/or bind DAxud1 in *Drosophila*. For this purpose, a TaDa-Seq experiment was performed. This technique is based on the activity of prokaryotic Dam methylase, which is fused to the transcription factor or chromatin element of interest. This chimera methylates GATC sites (recurrent sequence in genomes), making it possible to cut the fragments with restriction enzymes that recognize GAmeTC and ligate adapters to their cohesive ends [31]. The assay was performed using the protocol developed by Southall et al. [32], expressing the Dam-DAxud1 chimeric protein to identify the *loci* in which DAxud1 is located. This experiment was performed in third instar imaginal wing disc tissue using the *nub* > Gal4 driver, with one replicate per condition (with Dam as a control, and Dam-DAxud1 as an experimental condition forcing bias in DNA methylation to DAxud1 binding sites). The *findpeaks* script, generated by Marshall et al. [33], was employed to analyze the aligned sequences. With this experimental and computational method, 1903 significant peaks (or signal windows over background) were detected across the *Drosophila* genome. The information from these significant peaks was employed to generate a metagene profile and heat map occupancy with gene length as a reference scaled to 1000 bp (Figure 1A). The 1903 significant peaks were associated with genes and their surrounding regions. These peaks were arranged in 20% percentiles (five groups) according to their score (Figure 1A) and calculated by subtracting Dam-DAxud1 from the Dam control signal according to Marshall’s computational method [33]. Although there is a main tendency to locate in the promoter zone, transcription start site (TSS), and introns, there are some differences between percentiles: the first percentile with the highest scoring peaks has them distributed mainly in promoter zones and introns (Figure 1B), while the peaks from the lowest percentile have a lower percentage of peaks (Figure 1B); thus, the Dam-DAxud1 score is higher in zones near the genes and peaks, while lower scores are located in the intergenic zones. Note that, in the lower percentiles, a low signal is detected in intergenic regions, possibly due to their location in regions with heterochromatin. We performed a subsequent analysis of the annotated genes in each percentile and listed their classification in KEGG pathways (Table 1). This showed genes related to the Wnt pathway that are overrepresented in both the first and last percentiles. Similarly, there are categories such as apoptosis and the MAPK pathway. The associations of DAxud1 with Wnt and apoptosis have previously been described [21,24,25], while MAPK has not been documented before. However, MAPK is a pathway linked to cancer development, is a part of JNK signaling, and is required for stress transduction and mitogen signals [34], which is consistent with evidence that indicates that there is a link between DAxud1 orthologs and cancer.

### 2.2. Potential DAxud1 DNA Binding Motifs and Their Role in Regulatory Genomic Regions

After having found the main groups of genes in which Dam-DAxud1 is enriched in wing imaginal cells chromatin, our next task was to search for the most frequent binding motifs within the sequences in Dam-DAxud1 peaks. Those sequences had a size between 50–5000 bp and were extracted for *findMotifs* perl package from HOMER developed by Heinz et al. [35]. As shown in Figure 1C, the most recurrent DNA consensus motif (AYATACATAYATA) was present in 921 sequence peaks out of 1903 (48.4%). This sequence is different when compared with preceding reports on DAxud1 vertebrate orthologs [22,25], possibly due to the extensive coverage of the sequences that the TaDa-seq experiment provides [31,36], but this information could give insights into the chromatin context surrounding DAxud1 distribution. This consensus sequence was used as input for gene ontology for motifs (GOMo) [37], a tool that supplies information about the gene ontology (GO) associated with the promoter transcription start site (TSS) regions from the respective gene, using a range between −1000 and +200 bp from the TSS for each gene in the *Drosophila melanogaster* whole genome. For the motif found, the most closely related GO is “response to heat”, GO:009408, and “response to hypoxia”, GO:001666 (Figure 1C). These results suggest that DAxud1 may play a role in the regulation of heat shock and other stress-triggered genes. Dam-DAxud1 shows significant peak occupancy in *hsp* genes, including *hsp70Ba*, *hsp26*, and *hsp67*, with the presence of the motif in their promoter region (Figure 1D). These results prompt us to analyze the *loci* and expression of the *hsp* genes.

### 2.3. DAxud1 Knockdown Induces Increased Expression of Hsp70, Hsp67, and Hsp26, Whose Effect in Adults Is Greater Survival and Less Resistance to Heat Stress

The above results reveal the presence of DAxud1 in some *hsp* genes (*hsp70*, *hsp26*, and *hsp67*) and show that the main DNA binding motif (AYATACATAYATA) is associated with stress response as its presence in heat shock response genes, suggesting a possible role modulating stress response. This hypothesis is also supported by studies on Axud1 orthologs in mammals [27,29,30,38]. Even though “stress response” is not the main GO category of DAxud1 genome location (Appendix A), our next task was to test whether flies with a ubiquitous DAxud1 knockdown exhibit an abnormal stress response or aberrant *hsps* mRNA expression, due to its presence in genes related to heat shock response and the functional category of the putative binding motif found with TaDa-seq (Figure 1C). To do this, flies expressing a DAxud1 RNAi construct (Vienna stock 26479, *UAS-IR-dAxud1*) were used, and expression in all tissues was done with the Tubulin-Gal4 driver. Flies overexpressing DAxud1 (WT and C-terminal tagged) cannot be analyzed in this type of assay because ubiquitous expression of this protein is lethal due to its proapoptotic effect [24]. The confirmation of the efficiency of RNAi V26479 on *dAxud1* expression is documented in Appendix A, as well as the reversion of DAxu1 overexpression phenotype in imaginal wing discs documented in Appendix A. To enhance the expression of the UAS RNAi construct, adult flies were placed at 29 °C. After that, control and experimental adult flies were subjected to a half-hour daily heat shock at 37 °C as a thermal stress condition, and their survival was measured. This shows us that under the same heat stress condition, knockdown of DAxud1 resulted in a shorter lifespan of adult female flies as compared to the control genotype (Figure 2A,B). Surprisingly, in thermal control conditions (no heat shock), DAxud1 knockdown animals have a longer lifespan compared to control animals for both males and females (Figure 2A,B). To confirm whether there is a change in *hsp* gene expression due to *dAxud1* influence, and to connect this to the observed phenotypes, qPCR analysis was performed for *hsp70*, *hsp26*, and *hsp67* in imaginal wing discs and salivary glands, in knockdown (UAS-RNAi construct) or gain of function (expressing the UAS-DAxud1-GFP version), using the *nubbin*-Gal4 driver to express the UAS constructs. This makes it possible to study the effects of DAxud1 manipulations in salivary glands and imaginal wing discs, tissues that have cells in an endo-replication state (salivary glands) and in a mitotic or proliferative state (imaginal wing discs), also revealing whether there is a cell type-specific effect. Figure 2C–E shows the qPCR results for three *hsp* genes (*hsp70*, *hsp26*, and *hsp67*), with each bar representing the fold change relative to the control condition (nub > Gal4; +/+, only Gal4 expression at 29 °C). These experiments indicate that DAxud1 knockdown rises the expression of *hsp* genes in thermal control conditions (29 °C) but does not enhance the transcriptional heat shock response in either tissue. In addition, the increase of *hsp* mRNA expression at the control temperature can account for the longer lifespan, since there is a report in *Drosophila* that mild ubiquitous overexpression of *hsp* generates this effect [39]. Conversely, overexpression of DAxud1 reduces *hsp* basal expression at the control temperature, and even though it does not inhibit the transcriptional response to heat shock, this is of a lesser magnitude than the control or the knockdown of dAxud1 condition (Figure 2C–E). In addition, as was mentioned above, ubiquitous overexpression of DAxud1 generates lethality, so we only tested this condition in specific tissues; however, the DAxud1 RNAi effect in the expression of those genes in adult flies is documented in Appendix A, in which the increase of expression of these *hsp* genes is also observed. Given these findings, the following experiments were performed to clarify whether DAxud1 localizes to regulatory regions of the *hsp70* gene and whether this changes with the transcriptional response to heat shock.

### 2.4. DAxud1 Is Detected by ChIP in the Promoter of Hsp70, and Heat Shock Reduces Its Binding in Those Regions

Upon closer analysis of the qPCR data on *hsp* gene expression in DAxud1 knockdown conditions as shown in Figure 2; the results suggest that DAxud1 may regulate *hsp70* mRNA expression, potentially as a transcriptional repressor, but previous research has indicated that the DAxud1 ortholog in mice, CSRNP, possesses transcription factor features, including a trans-activating domain at the C-terminus and a DNA binding motif [22]. Nonetheless, there is no concrete evidence that sheds light on the nature of the regulatory factor that DAxud1 could be, its dynamics within the chromatin, and the elements of the structure it interacts with. Due to the unavailability of antibodies against DAxud1, we employed a GFP-tagged version of DAxud1 (DAxud1-GFP), which replicates the phenotype observed when endogenous DAxud1 is overexpressed [24], allowing us to perform immunoprecipitation using an anti-GFP antibody. With chromatin immunoprecipitation (ChIP) for DAxud1-GFP, we detected the presence of this factor by measuring the enrichment with qPCR (ChIP-qPCR), using primers to the *hsp70* paralogs *hsp70Aa* and *hsp70Ba.* We also looked at the *Drosophila Hsp70Aa* paralogs, as their promoters and non-coding elements have many similarities with *the hsp70A* and *hsp70B* paralogs. We used a promoter, TSS zones, and an additional pair of primers to screen the TaDa-seq significant peak zone in −2 kb of *hsp70Ba*, in which there is an AYATACATAY motif as well. According to these results (Figure 3), DAxud1-GFP has a clear presence in both *hsp70Aa* and *Ba* promoters. Additionally, ChIP-qPCR was performed for a region in which the Dam-DAxud1 signal had a signal close to zero or negative, such as a promoter region of the *Oct3Rbeta locus* (Figure 3, right side), which is near the *hsp70B* cluster, confirming the TaDa-Seq’s results and showing the background signal from non-occupied (or non-significant peak) regions. After heat shock, the DAxud1 signal decreases in promoter zones, and this effect is noticeable in the intergenic (or enhancer) area where the AYATACATAYATA motif found by TaDa-seq is located (Figure 3).

### 2.5. Changes in DAxud1 Levels Affect the Status and Positioning of RNA Polymerase II, Suggesting a Repressive Role

The above results lead us to propose that DAxud1 exerts a repressive function over *hsp70* expression. Within the *hsp70* genes, a distinct chromatin configuration exists called the pausing complex. This complex is characterized by the RNA Polymerase II which pauses its elongation at positions +20 to +50 from TSS, which enables quick release and elongation upon the application of a heat shock as a release signal [4]. The poised polymerase involves two main components different from canonical transcription factors, NELF and DSIF, which work together to maintain the polymerase in a paused state. To resume transcription, the p-TEFb complex phosphorylates the RNA Polymerase II, NELF and DSIF, releasing the active polymerase leading to an elongation [4]. The conditions in which the RNA Polymerase remains poised reside in chromatin configuration and elements that facilitate the phosphorylation of the pausing complex due to TEFb kinase activity [4]. Considering how the pausing release process works and the results from the DAxud1 analysis, we conjecture that DAxud1 could influence *hsp70* expression, counteracting the transition to the elongation stage.

To examine this hypothesis, we tested if DAxud1 could influence the position of RNA Polymerase II around the *hsp70* gene, analyzing the distribution in the 5′ and 3′ regions of the gene. We conducted chromatin immunoprecipitation (ChIP) for non-phosphorylated RNA Polymerase II in salivary glands under varying levels of DAxud1 expression and heat shock induction (Figure 4A,B). Our findings suggest that in the case of DAxud1 knockdown, there is a uniform decrease in RNA Polymerase II levels across the *hsp70* gene *locus*, without any change in the 5′/3′ rate of RNA Pol II compared to the control (Figure 4C,D). In contrast, DAxud1 overexpression generates an increase in this ratio, more so than in the control condition, due to an increase of the RNA Pol II signal at the 5′ end of the *hsp70* gene body. This result is noteworthy since we used an antibody against the non-phosphorylated C-Terminal Domain (CTD) of the RNA Polymerase II [40]. Therefore, more than the position, it could be influencing the phosphorylation state.

In summary, the results of the distribution of DAxud1 in the *hsp70 locus*, the effects of its manipulations on the expression of *hsp70*, and its influence on the positioning of RNA Polymerase II suggest a repressive role of DAxud1 on *hsp70* transcription and possibly other genes related to stress response and survival.

## 3. Discussion

### 3.1. Role of DAxud1 in Tissue Homeostasis through Hsp Regulation

Finding a recurrent binding site in promoters from the *hsp* gene family between the most recurrent genomic zones where DAxud1 is positioned in the *Drosophila* genome prompts us to perform thermotolerance assays, and thus test the physiological relevance of this gene under this stressful condition. The expressions of three genes, *hsp70*, *hsp26*, and *hsp67*, were analyzed during knockdown and overexpression of DAxud1 to establish if there are any links between the expressions of these genes (Figure 2C–E). With respect to physiological relevance, tested with lifespan assays in conditions of daily heat shock, the knockdown animals showed reduced thermal resistance compared to the same control genetic background with no heat shock. In parallel, the DAxud1 knockdowns had a higher life expectancy even compared to the genetic control condition, if they were not subjected to heat shock (Figure 2A,B). It is noteworthy that DAxud1 knockdown larvae raised at 29 °C (no heat shock) displayed higher levels of *hsp70, hsp26*, *and hsp67* expression in larvae salivary glands, imaginal wing discs, and the whole adult compared to the control animals kept at the same temperature (Figure 2C–E and Appendix A). In *Drosophila*, the effect of *hsp* overexpression is well documented, and, precisely, one of its effects is an extension in lifespan due to its cytoprotective and anti-apoptotic properties [39,41,42], explaining why DAxud1 knockdown extends lifespan at the control growing temperature (Figure 2A,B). On the other hand, when animals with reduced DAxud1 levels are exposed to daily heat shock stress, their lifespan shortens, which did not occur with the control animals, in which the lifespan remained within the same range, without exhibiting a decrease (Figure 2A,B). However, the diminished tolerance to stress of DAxud1 knockdown animals could be explained by two complementary mechanisms: the first mechanism is related to the JNK pathway and apoptosis induction because it is known that DAxud1 has a pro-apoptotic role through the activation of JNK signaling [24,29]. Therefore, when considering the impact of DAxud1 knockdown, it is possible that the activation of the JNK signaling is decreased. This could potentially result in the survival of damaged cells and disruption of regenerative processes, which are known outcomes of impaired JNK signaling according to previous research [43,44,45]. This hypothesis is reinforced by the presence of Dam-DAxud1 on the *Loci* of genes associated with the apoptotic pathway linked to the JNK pathway, in which DAxud1 may have an activator role (Table 1, Appendix A). Considering the effects of heat shock on signaling pathways and tissue homeostasis, it is worth noting that another study reports that, after a prolonged heat shock (29 °C for 8 days), *Drosophila* fat body cells enter apoptosis [46]. This means that the heat shock response, after some time, is overwhelmed by cell damage and apoptosis signaling takes place. In additional experiments in YW larvae, *dAxud1* mRNA levels do not increase significantly during acute heat shock (37 °C for 30 min), but their levels do increase significantly with long-term heat stress (32 °C for 72 h), under which DAxud1 could exert its pro-apoptotic role (Appendix A). This rise in expression under long-term stress has been documented in homologues of DAxud1 when tissues are exposed to cerium [26] or Doxorubicin [27] and has been shown to coincide precisely with apoptosis. The second mechanism is related to the overexpression of *hsp70* itself: in *Drosophila*, the overexpression of high levels of *hsp70*, at higher levels than heat shock induction, is detrimental for cells and larvae because the biosynthesis of Hsp70 protein has a high translation rate, consuming nitrogen resources [20,47], so this situation could harm the regeneration process. Therefore, DAxud1 could have a role not only in promoting apoptosis [24] and neural differentiation [25], but also in keeping the levels of other proteins that interfere with protein development and homeostasis at bay, thus being a protein with a pleiotropic role.

Regarding this point, the two proposed mechanisms converge in the documented role of DAxud1 and their homologues from the CSRNP family as tumor suppressors [21,24], in which low levels of expression are associated with poor prognosis cancers [48]. This may not only be due to decreased pro-apoptotic activity resulting from lower Axud1 levels in mammals, but also to increased *hsp70* expression at levels that foster cell survival. Likewise, this relation (lower Axud1 expression and higher Hsp70 levels) has been independently documented in multiple studies, being related to progression and poor prognosis for prostate cancer [49,50], lung squamous cell carcinoma [19,51], and hepatocellular carcinoma [14,52]. In any case, this relationship between low levels of Axud1 and high levels of Hsp70 in the progression and aggressiveness of cancers must be confirmed in studies conducted for this purpose.

### 3.2. How Does DAxud1 Exert Its Repressive Effect?

Although *dAxud1* expression levels do not change in a significant way during acute heat shock as 37 °C for 20’ (Appendix A), when this condition is induced, the protein DAxud1 moves out the promoter and regulatory zones of *hsp70* (Figure 3). The changes of DAxud1 location in the chromatin, observed in Figure 3, can be mediated by putative phosphorylation sites or by the oxidation state of the cysteine-rich region present in DAxud1 [22,24], inducing conformational changes that could modify its affinity for DNA or other elements of chromatin. Studies have demonstrated that the cysteine-rich can function as regulators of protein conformation, wherein heat or oxidative stress can modify the disulfide bonds, changing protein conformation. This phenomenon has been well-documented in proteins that possess cysteine-rich regions [53,54]. Consequently, alterations in DAxud1 protein conformation induced by post-translational modifications could potentially be a crucial factor in regulating its localization in *hsp70 loci* under normal conditions and its removal during acute heat shock to foster the heat shock response, which exerts its cytoprotective effect in the cell through the inhibition of apoptosis and the conservation of protein folding during acute heat shock [55,56].

The conformational changes discussed above could be an intrinsic property of DAxud1 to modulate gene expression; however, with respect to the chromatin context, *hsp70* TSS has a chromatin arrangement known as the pausing complex, in which the RNA Polymerase II stops its elongation stage at positions +20 to +50 bp from the TSS and remains stalled as a “poised polymerase”, allowing a rapid release and elongation upon heat shock as a release signal [57]. This poised polymerase makes a complex with two main components, NELF (composed of four Nelf proteins) and DSIF (composed of Spt5 and Spt6), whose role is to maintain the polymerase in this paused state [58]. Transcription resumes when the Heat Shock Factor (HSF), induced by a rise in temperature, changes its conformation and places itself in the promoter zone, targeting the positive Transcription Elongation Factor b (p-TEFb complex) on the pausing complex, phosphorylating the NELF and DSIF complexes [59]. The phosphorylated NELF complex disassociates from RNA Polymerase II, while the phosphorylated DSIF complex goes from being an element that maintains pausing to being an elongation factor that is maintained with the active RNA Polymerase II in the elongation stage [4]. In addition, p-TEFb phosphorylates the C-Terminal Domain (CTD) from RNA Polymerase II, turning it into the transcriptionally active form (also named RNA Polymerase IIo) [60], with this kinase activity being complementary to the general transcription factor TFIIH, which phosphorylates the CTD to initiate the transcription in most genes [61]. It should be noted that this phosphorylated form is also the one that transcribes other genes of the *hsp* family (*hsp26*, *hsp67Bc*) during heat shock, being documented even in polytene chromosomes, whose presence in those *loci* was detected by microscopy [62]. This may shed light on the results of the RNA Polymerase II ChIP, which was performed with the 8WG16 antibody (Figure 4A,B), whose epitope is the unphosphorylated heptapeptide of the CTD domain of RNA Polymerase [40]. Although the % INPUT 5′/3′ ratio remains similar to the control (Figure 4C,D) under DAxud1 knockdown conditions, with a uniform decrease in signal throughout the gene body, this may be due to a greater proportion of phosphorylated CTD (RNA Polymerase IIo), which is not the epitope target of this antibody. This biased enrichment of the TSS means that the unphosphorylated isoform is located predominantly in the transcription preinitiation or pausing complexes. In this context, it would be of interest to elucidate whether DAxud1 intervenes at the level of TFIIH or p-TEFb, or if it interacts with the other pausing complexes. With respect to the latter hypothesis, although it was not possible to verify it in this study, a report on two-hybrid experiments suggests a physical interaction between DAxud1 and the DSIF subunit Spt5 [63], meaning that DAxud1 could pause induction in *hsp70*; however, these hypotheses must be tested with appropriate experiments in the future.

Another possible mechanism may be based on the physical occupation of the regions where the binding sites of other proteins are located, which are necessary to recruit activating elements such as HSF, BRD4, and p300, which play a role in how p-TEFb is located over the pausing complex to phosphorylate DSIF, NELF, and RNA Polymerase II [4]. Although the TaDa-seq data indicate that DAxud1 is present in an intergenic area of *hsp70Ba*, and *hsp26*/*hsp67Bc* promoters (Figure 1C,D) and later confirmed by ChIP (Figure 3), up to date, no detailed studies have been conducted on the protein domains of DAxud1 or its homologues, which could provide information on interactions or other specific enzymatic activities associated with histone modification on those regions that can determine a repressive or activating role, so this remains a possibility to study.

In summary, these results lead us to conclude that DAxud1, in addition to a transactivating role in transcription that was previously described in its vertebrate homologues [22,25], also has a repressive role. The mechanisms of how this effect occurs must be elucidated, but in this discussion, multiple mechanisms have been considered that could explain this effect, and, more importantly, answer the question about the timing of DAxud1 activity after stress, because the cell has a first stress response to survival mediated by *hsp70* and other elements, and this response is eventually overwhelmed and the apoptosis mechanism prevails in order to preserve tissue integrity.

## 4. Materials and Methods

### 4.1. Fly Stocks

For DamID-seq assay (TaDa-seq), the genotypes were *nub* > Gal4/Y; *UAS-mCherry.Dam* and *nub* > *Gal4*; *UAS-mCherry.Dam::dAxud1*, both were generated from the plasmid pUAST-attb-LT3-Dam provided by the Andrea Brand Laboratory. Control flies (*UAS-mCherry*) were generated with the plasmid *pUAST-attB-LT3-Dam* in eggs from flies *P{y[+t7.7] = CaryP}attP2* (BL8622 stock, from BDSC, IN, USA). *UAS-mCherry.Dam::dAxud1 stock* was generated by injecting the plasmid *pUAST-attB-LT3-Dam-dAxud1* in this fly stock. The construct was generated by cloning the *dAxud1* gene from the plasmid *pUAS-dAxud1::GFP* [24] using the primers dAxud1-notI-FW (5′-ACCAGCGGCCGCGATCTAGGGACCTCGACACGGATAGCATGT-3′) and dAxud1-XbaI-Rv (5′-ATATGGTCTAGATTATTAACCGGTGGATCCCGGGCCCGCGGGGAGG-3′). To induce loss of function or knockdown, *dAxud1* RNAi was expressed via the GAL4/UAS system utilizing the Vienna stock V26479 (Vienna Drosophila Resource Center (VDRC), Vienna, Austria).

### 4.2. DAxud1 TaDa-Seq

In sum, 100 to 120 imaginal wing discs were dissected from third instar larvae (*nub > Gal4*; *UAS-mCherry.Dam* (control) and *nub > Gal4*; *UAS-mCherry.Dam::dAxud1* (experimental)). After growing for 7 days at 17 °C, the larvae were subjected to 29 °C for 24 h, reaching to the third instar stage. Following dissection in PBS1× at 4 °C, the DNA was extracted according to the Southall protocol [32]. The methylated and processed DNA was used to generate libraries and sequenced in Illumina Hiseq−2500, single end 100 bp. The data were processed with the pipeline designed and published by Marshall [33] using the dm3 *Drosophila* genome version (release 5.57). The data from Dam and Dam-DAxud1 were analyzed to generate a bedgraph data file, according to the pipeline. From these results, a GFF file with the coordinates of significant peaks was visualized with the IGV epigenome browser. These data were processed with HOMER perl package *annotatePeaks* [35], XTREME was used for motif discovery and sequence peak alignment, and GOMO was used to perform promoter gene ontology, both programs available in MEME suite developed by Bailey et al. [37]. The profiles (heatmap and metagene plots) were generated with the *deeptools* toolbox using the packages *ComputeMatrix* and *PlotHeatmap*, all developed by Ramírez et al. [64].

### 4.3. Chromatin Immunoprecipitatin (ChIP-PCR)

ChIP was performed with salivary glands material expressing DAxud1-GFP or *dAxud1* RNAi growth at 25 °C using the driver *nubbin* > Gal4. Larvae from the heat shock condition were exposed to 37 °C for twenty minutes. Salivary glands were dissected, and the tissue and cells were processed according to Ghosh et al.’s procedure with three biological replicates per experimental condition [65]. Briefly, ten pairs of salivary glands from third-instar larvae were incubated for five minutes on ice in 100 μL of 1% formaldehyde in PBS 1× and then at RT for seven minutes. Quenching to stop the crosslinking was achieved by adding 2.5 M glycine to the fixation mix to a final concentration of 125 mM; after this addition, the glands were placed on ice for two minutes. The fixed glands were precipitated and centrifuged at 900× *g* for 2 min at 4 °C, and the supernatant was removed. Additionally, 100 μL of sonication buffer (20 mM Tris (pH 8.0), 0.5% SDS, 2 mM EDTA, 0.5 mM EGTA, 0.5 mM phenylmethylsulphonyl fluoride (PMSF)) and 1 μL protease inhibitor cocktail (number cat. 78430, Thermo Fisher Inc., WLM, USA) were added to the glands to incubate at room temperature for ten minutes followed by another ten minutes incubation in ice. The glands were shaken in a vortex (medium speed) for ten minutes and homogenized with a small polypropylene pestle for further sonication at 4 °C in Omniruptor-4000 (Omni International, Kennesaw GA, USA) at 100% power and 90% pulse for fifteen minutes to shear the DNA to a fragment size of 400 bp as average. The lysate was cleared from debris by centrifugation at 14,000× *g* for 7 min. In total, 45 μL per lysate was used for each immunoprecipitation assay (mock and IP), with 10 μL for input. Immunoprecipitation was performed using 1 μL of anti-GFP antibody (ab290, Abcam, Cambridge, UK), 5 µg for chromatin from ten pairs of salivary glands) or 30 μL of RNA Polymerase II antibody directed to the unphosphorylated heptapeptide repetition domain in CTD 8WG16 (cat. sc-56767, Santa Cruz Biotechnology, Paso Robles, CA, USA), 6 µg for chromatin from ten pairs of salivary glands), diluted to half its concentration in IP Buffer (50 mM Tris-HCl pH 8; 100 mM NaCl; 2 mM EDTA; 1 mM EDTA, 1% NP40 1 × halt protease inhibitor cocktail (cat. 78429, Thermo Fisher Inc., WLM, Waltham, MA, USA)), and immunoprecipitated with 30 μL of protein-A Dynabeads^®^ (cat 10001D, Thermo Fisher Inc., WLM, Waltham, MA, USA) with the same amount for the mock experiment (only beads). Beads were washed twice in sequence with ChIP 1 Buffer (IP buffer + 1% Na-deoxycholate), ChIP 2 Buffer (IP buffer + 1% Na-deoxycholate + 500 mM NaCl), and ChIP 3 Buffer (IP buffer + 1% Na-deoxycholate + 270 mM LiCl); then, the beads were washed twice in Tris-HCl 10 mM pH 8. Finally, elution buffer (NaHCO_3_ pH 8.8 0.1 M; SDS 1%), was added to the precipitated beads until 200 μL was reached (same as input). Beads were isolated from the eluted material with magnet.

Decrosslinking was performed by adding 10 μL of NaCl 4 M and 0.5 μL of fungal proteinase K (cat. 25530015, Thermo Fisher Inc., WLM, USA), with six hours of incubation at 65 °C. The samples were purified and eluted with MicroChip Diapure Columns (cat. C03040001, Diagenode, Ougrée, Belgium). Samples were analyzed with qPCR. The primers used were as follows: *hsp70B* paralogs (+26, +131) Fw 5′- GCTAAGCAAATAAACAAGCGCAG-3′, Rv 5′-CAGTTGATTTACTTGGTTGCTGGT-3′; *hsp70B* promoter (−201, +52) Fw 5′-TTCTCTGGCCGTTATTCGTT-3′, Rv 5′-TCGAACCAACGAGAGCAGTA-3′; *hsp70Ba* TSS (−113, +131): FW 5′- CGACATACTGCTCTCGTTGG-3′, RV 5′-CAGTTGATTTACTTGGTTGCTGGT-5′; and *hsp70Ba* 3′ region (+1659, +1859) Fw 5′-AATGGAATCCTGAACGTCAGC-3′, Rv 5′-TACGTTGTATACGTAGCTCTCCAG-3′. Additional primers for *hsp70A* paralogs were extracted from reference to span the TSS 5′ and 3′ regions, and were named CHIP-A and CHIP-B, respectively [66].

### 4.4. RNA Extraction and qPCR

The TriZol reagent (cat. 15596026, Thermo Fisher Inc., WLM, USA) was used to extract RNA, in accordance with the manufacturer’s instructions. After resuspending the RNA pellet in water and measure concentration, the iScript^®^ kit (cat. 1708841, Bio-Rad, CA, USA) was used to perform the retrotranscription reaction. The qPCR mix reactions were conducted using Billiant SYBR^®^ (cat 600828, Agilent, Santa Clara, CA, USA), with an initial denaturation step of 10 min and 40 cycles using the following sequence: 30 s at 90 °C, 30 s at 60 °C, 30 s at 72 °C. The primers employed in this study were designed as follows: *dAxud1* Fw 5′-AGGGGACCACCAGCCTAAC-3′, Rv 5′-GGTTCGCTCTGATTATCCTTGTG-3′; *hsp26* Fw 5′-ATGCCCACGATCTGTTCCATC-3′, Rv 5′-GTACGCGAATAACGACGAC; *hsp67Ba* Fw 5′- TCATCCGTACAATCGTGTGGC, Rv 5′-CGTGCCCTTATCGCCCAAT-3′; *hsp70B* 3′ Fw 5′-GAGGATTTGGCGGCTACTCT-3′, Rv 5′-TTTAAAAACTTAAGCCAGGAACTGA-3′; and *actin-42A* (as normalizer) Fw 5′-GCGTCGGTCAATTCAATCTT-3′, Rv 5′-AAGCTGCAACCTCTTCGTCA-3′.

### 4.5. Lifespan Assays

Each replicate assay involved the collection of 80–100 flies, which were gender separated within 48 h of pupae eclosion. Three replicates were conducted per condition, and both living and deceased flies were tallied daily. In the control condition, the flies were moved to new tubes every 2 days and were maintained at 29 °C. The heat shock condition involved exposing the flies to 37 °C for 30 min every day.

## Figures and Tables

**Figure 1 ijms-24-07485-f001:**
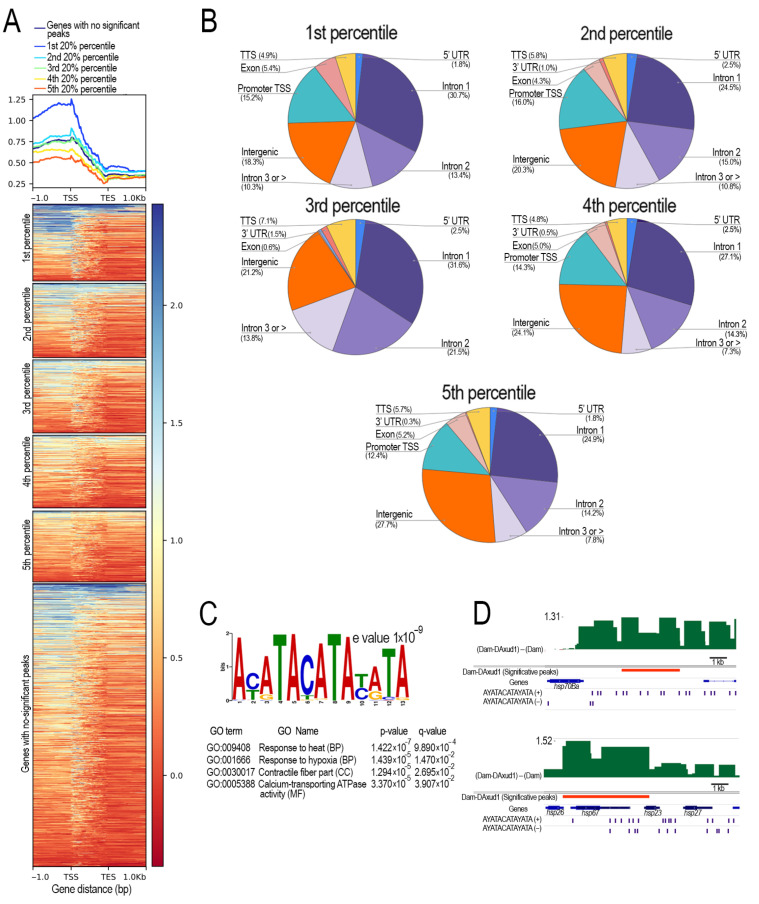
TaDa-seq results from Dam-DAxud1/Dam normalization show the genome-wide location and main motif in the identified significant binding zones. With the Dam/DAxud1 pipeline, a total of 1903 peaks were identified as significant over the background via signal subtraction (Dam-DAxud1 and Dam (no fusion)). These peaks were then annotated with HOMER. (**A**) In the upper zone, a metagene profile is depicted, with Dam-DAxud1 signal (subtraction) with all genes scaled to 1 kilobase. The profile represents the average of the signal every 10 bp bin size, with colors according to their percentile, from the first (1st) with the highest enrichment to the last (5th) for all genes with significant peaks surrounding its region (−1 kb TSS to + 1 kb TTS). Gene regions without significant regions are in dark blue. At the bottom of this figure are the heat maps for genes with regions that have statistically significant peaks, grouped into five percentiles. In the lower block, genes that do not have significant peaks in their surroundings were plotted. (**B**) Location of peaks in genomic regions grouped by percentiles. (**C**) Consensus and most recurrent sequence in the peaks of the DAxud1 TaDa-seq experiment. The Xtreme tool (MEME suite) was used to analyze the sequences. Below, the GOMO tool (MEME suite) was used to explore the relationship between this consensus sequence and the promoters in which it is found, according to gene ontology classifications, based on which it is deduced that it may be present in promoters of heat shock genes. (**D**) Genomic context of heat shock gene promoters, whose regions coincide with significant Dam-DAxud peaks. The Dam-DAxud1 signal from signal subtraction is illustrated in green; in red, the region significantly enriched and identified by the “Find-Peaks” analysis tool; and, in blue, the coding regions of these genes. Below are denoted the regions with the identified consensus sequences.

**Figure 2 ijms-24-07485-f002:**
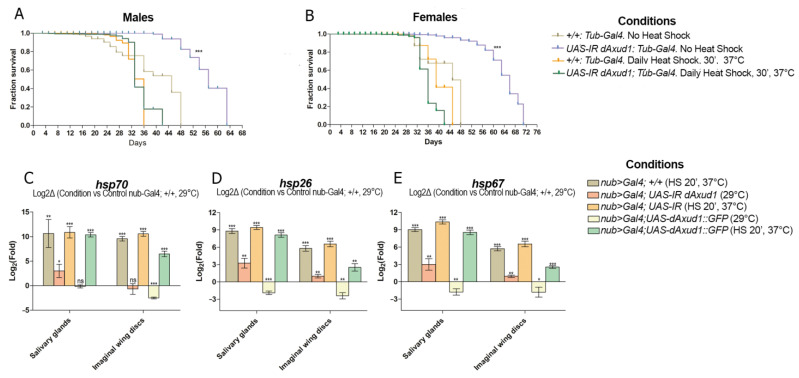
The impact of *dAxud1* knockdown in lifespan. (**A**,**B**) represent the survival chart to depict the effect of *dAxud1* knockdown (*yw; UAS-IRdAxud1/+; Tub > Gal4/+*) vs. control genotype (*yw; +/+; Tub > Gal4/+)* when they are exposed to heat shock episodes of 37 °C for 30 min every day. NHS: non heat shock; DHS: daily heat shock. (**C**–**E**) represent the relative expression of three heat shock genes, *hsp70B* (all paralogs), *hsp26*, and *hsp67*, in two different tissues. The expression was analyzed on five conditions and compared to control conditions, including *dAxud* knockdown (*nubbin* Gal4/+; *UAS-IR-dAxud1*) and DAxud1 overexpression (*nubbin* Gal4/+; UAS-DAxud1-GFP). The control genotype was *nubbin* Gal4/+ (driver only). All bars represent the Log2Fold change of the condition compared with the control both at genotype (*yw; +/+; Tub > Gal4/+*) and temperature (29 °C). Significance from comparative analyses (condition vs. control) is based on a *t*-test with * for *p* < 0.05, ** for *p* < 0.01, and *** for *p* < 0.0001. ns: statistically non-significant.

**Figure 3 ijms-24-07485-f003:**
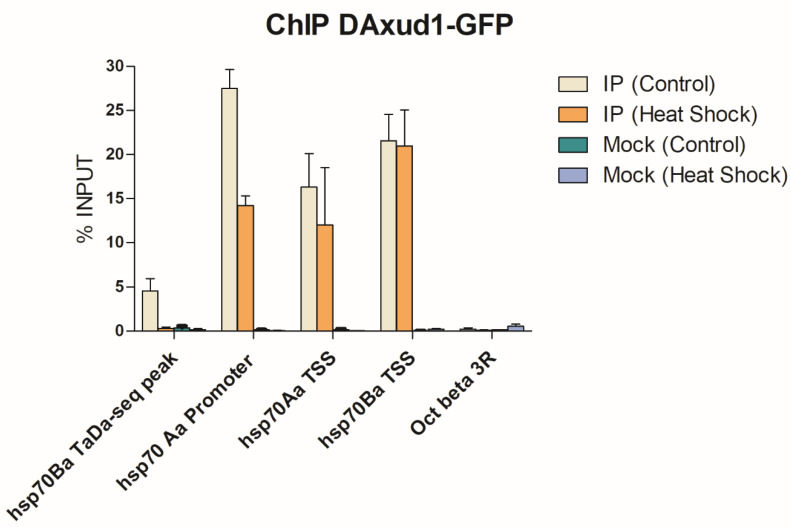
DAxud1-GFP chromatin immunoprecipitation (ChIP). Assays for different *loci* were performed under control conditions (25 °C) and heat shock (37 °C for 20 min). The mock consisted of chromatin immunoprecipitation with Dynabeads only.

**Figure 4 ijms-24-07485-f004:**
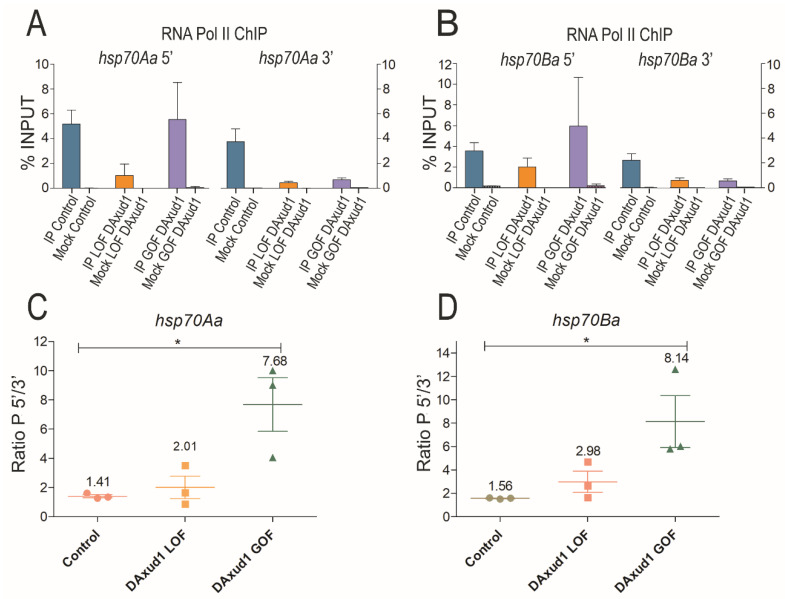
Unphosphorylated RNA Polymerase II chromatin immunoprecipitation (ChIP). Assays were performed in control conditions of 25 °C, loss of function by RNAi knockdown (LOF), and gain of function overexpressing DAxud1-GFP (GOF). The ChIPs were performed for TSS-5′ and -3′ regions of *hsp70Aa* (**A**) and *hsp70Ba* (**B**). The % input from the RNA Polymerase IP was used to obtain a ratio of 5′ and 3′ for *hsp70Aa* (**C**) and *hsp70Ba* (**D**) in which mean is above the groups. One way ANOVA analysis was performed in groups from panels C and D. * for *p* value < 0.05.

**Table 1 ijms-24-07485-t001:** KEGG pathways from TaDa-seq annotation genes with significative peaks grouped into five percentiles.

**1st percentile**	**Term**	**Genes in Term**	**% Enrichment**	***p*-Value**	**FDR**
	dme04310:Wnt signaling pathway	9	2.586206897	0.0009865	0.02224402
	dme04391:Hippo signaling pathway—fly	6	1.724137931	0.00437593	0.12252595
	dme04013:MAPK signaling pathway—fly	7	2.011494253	0.00901989	0.16837131
	dme04068:FoxO signaling pathway	5	1.436781609	0.03730033	0.52220455
**2nd percentile**	dme04391:Hippo signaling pathway—fly	6	1.719197708	0.00355027	0.18106386
	dme00230:Purine metabolism	6	1.719197708	0.0274048	0.40818985
	dme04013:MAPK signaling pathway—fly	6	1.719197708	0.02855552	0.40818985
	dme04068:FoxO signaling pathway	5	1.432664756	0.03201489	0.40818985
**3rd percentile**	dme04013:MAPK signaling pathway—fly	10	2.915451895	7.1911 × 10^−5^	0.00264315
	dme04391:Hippo signaling pathway—fly	8	2.332361516	9.6114 × 10^−5^	0.00264315
	dme04214:Apoptosis—fly	7	2.040816327	0.00137923	0.02528582
	dme04392:Hippo signaling pathway—multiple species	3	0.874635569	0.03916055	0.53845762
**4th percentile**	dme04392:Hippo signaling pathway—multiple species	4	1.162790698	0.00469003	0.3470621
	dme04341:Hedgehog signaling pathway—fly	5	1.453488372	0.01092496	0.40422338
	dme01212:Fatty acid metabolism	5	1.453488372	0.02826409	0.47347768
	dme04213:Longevity regulating pathway—multiple species	5	1.453488372	0.03185866	0.47347768
**5th percentile**	dme04310:Wnt signaling pathway	10	3.03030303	3.0041 × 10^−5^	0.00132182
	dme04013:MAPK signaling pathway—fly	9	2.727272727	7.466 × 10^−5^	0.00164252
	dme04391:Hippo signaling pathway—fly	6	1.818181818	0.00159687	0.02342078
	dme04350:TGF-beta signaling pathway	5	1.515151515	0.00416474	0.03690422

## Data Availability

Illumina sequencing raw data from TaDa-seq is available in NLM with the identifier PRJNA776616.

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
