# Peer review of "Drosophila DAxud1 Has a Repressive Transcription Activity on Hsp70 and Other Heat Shock Genes"

_ijms, 2023, doi:10.3390/ijms24087485_

Round 1

Reviewer 2 Report

Proposed manuscript is aimed to describe DAxud1 as a new Drosophila transcriptional regulator. The manuscript provides significant amount of important information regarding roles of DAxud1 in controlling transcription of stress-response genes and its influence on Drosophila lifespan. But I suppose the manuscript needs a thorough rewriting, especially concerning proposed mechanisms of DAxud1 action. Many of the statements should to be smoothed or completely removed. Also, the manuscript would benefit from a few additional experiments.  

Major concerns:

1)     The most important idea which I suggest to be re-discussed is the description of the DAxud1 as a pausing factor.

In fact, evidences which support this description are rather insufficient. The authors made the link between DAxud1 and pausing NELF and DSIF complexes basing on dynamics of these complexes at the hsp70 locus upon its heat-induced activation. But in fact, there are quite a lot of differences in their behavior:

a)      According to Fig3D, DAxud1 binds upstream part of the hsp70 promoter in uninduced state, while Pol II peak is shifted towards coding region of the gene (and NELF and DSIF peaks which recruited through the binding of PolII)

b)     According to Fig3D, DAxud1 level decreases upon heat shock. While it is generally accepted that NELF and especially DSIF do not much decrease at the promoters upon heat stress. Indeed, NELF dissociated of the Pol II complex during its movement into gene bodies upon transcriptional induction, but NELF and DSIF still continue to be recruited at the promoters. Indeed, for the NELF was shown some decrease upon heat stress at the hsp70 promoter (https://www.nature.com/articles/nature08449), but it still has a significant level of binding as well as DSIF (https://www.sciencedirect.com/science/article/pii/S1097276503005264).

And that is important the loss of NELF and DSIF upon transcriptional induction is not the general mechanism of their action on Drosophila genes (https://www.nature.com/articles/s41598-020-80650-1).

Rather NELF and DSIF decrease at the promoters is a specific feature of transcriptional regulation of heat-induced genes.

So, dynamic of DAxud1 at the hsp70 promoter does not leas to the conclusion that DAxud1 works like NELF and DSIF

c)      DAxud1 influence on the transcription of heat-induced genes does not similar to NELF and DSIF action. Indeed, its knockdown leads to the significant increase in hsp70 transcription (which in fact not the case of NELF and DSIF influence on the hsp70 transcription – NELF knockdown leads to a much less increase in hsp70 transcription – hsp70 transcription increases 3-times but not 8-times http://genesdev.cshlp.org/content/22/14/1921.short). That is more – overexpression of DAxud1 leads to the strong decrease in heat-induced level of heat-shock genes, which was not demonstrated for the NELF action.

DAxud1 influence of heat shock genes transcription leads to the conclusion that DAxud1 is rather a repressor of heat shock genes that controller of their transcription by pausing mechanisms.

2) The submitted manuscript would benefit if all submitted experimental procedures were submitted in a coordinated manner:

a) ChIPs Fig3D and Fig4B should be analyzed using the same primers set.

b) Polythene chromosomes staining Fig3 should be performed both in control and heat-stress conditions to support the statement of DAxude1 relocation from the hsp70 loci upon heat stress and to confirm Fig3D ChIP result.

c) Please provide the background ChIP signal level (at some negative control site) for all the ChIPs. This is of crucial importance as let to analyze if DAxud1 level decrease at the hsp70 locus is simply a decrease or its complete removal.

d) It would be a good addition to analyze all the ChIPs at other heat-induced loci (hsp26 and hsp67)

Minor concerns:

1)     The manuscript text needs a substantial proofreading. All the scientific terms should be carefully verified (examples of wrong terms – double-hybrid assay, hsps genes..)

2)     Please specify at Fig2C-E between which samples you have calculated the statistics

3)     Lines 193-197 – according to abcam site your Pol II Ser5P antibodies do not recognizes unphosphorylated form of Pol II. Please correct this sentence.

4)     Fig.1 – Please magnify all the comments at the figure

5)     Please propose in the Discussion the possible explanation why you detect both the increase in hsp70 transcription upon DAxud1 depletion (Fig 2) and at the same way the decrease in Pol II binding level both at 5’ and 3’ regions of hsp70 gene.

Round 2

Reviewer 1 Report

1. The figure 3 shows after figure 4, please rearrange the panel.

2. I am still not convinced that only ten pairs of salivary glands can generate reliable results of ChIP experiments, can authors provide some successful examples?

Author Response

1. The figure 3 shows after figure 4, please rearrange the panel.

Fixed.

2. I am still not convinced that only ten pairs of salivary glands can generate reliable results of ChIP experiments, can authors provide some successful examples?

We fully understand your concern and we tested if that material was enough. Because some references used over 200 pairs of salivary glands, and they dilute at 1/10 to perform the IP.

The reference in the methods (Gosh et al,. 2011, https://doi.org/10.1128/MCB.05930-11), indicates that ten pairs is sufficient, and only 10 microliters from the inicial lysate works for the IP, as is described:

We considered that could be not enough and we used 45 microliters for our IPs and works well. The reference was added to that method section so that other researchers have an additional support on this way of doing the IP of salivary glands.

Additionally, there are other studies in which similar amounts of salivary glands are used for chromatin immunoprecipitation: Posukh et al., 2017 (https://doi.org/10.1186/s13072-017-0163-z) used 25 pairs of salivary glands for ChIP-seq (It is necessary to highlight that in NGS such as Chip-seq, the amplification of the libraries only reaches to cycle 13, to avoid sequencing biases).

Also, Legube et al., 2006 used ten pairs of salivary glands too (https://doi.org/10.1101%2Fgad.377506).

We hope that clears your doubts. And we thank you in advance for your review.

Reviewer 2 Report

I would recommend authors to carefully check the text before final publication (correct typos such as KEEG -> KEGG)

Author Response

Thanks for all your comments and review time.

We  will upload the revised manuscript again considering these abbreviations, once the platform opens again for resubmission.